# Hydrothermal Co-Processing of Coal Fly Ash Cenospheres and Soluble Sr(II) as Environmentally Sustainable Approach to Sr-90 Immobilization in a Mineral-like Form

**DOI:** 10.3390/ma14195586

**Published:** 2021-09-26

**Authors:** Tatiana Vereshchagina, Ekaterina Kutikhina, Leonid Solovyov, Sergei Vereshchagin, Elena Mazurova, Alexander Anshits

**Affiliations:** 1Federal Research Center “Krasnoyarsk Science Center of Siberian Branch of the Russian Academy of Sciences”, Institute of Chemistry and Chemical Technology, 50/24 Akademgorodok, 660036 Krasnoyarsk, Russia; ekaterina_kutikhina@mail.ru (E.K.); leosol@icct.ru (L.S.); snv@icct.ru (S.V.); len.mazurowa@yandex.ru (E.M.); anshits@icct.ru (A.A.); 2Department of Chemistry, Siberian Federal University, Svobodny Av. 79, 660041 Krasnoyarsk, Russia

**Keywords:** cenospheres, Sr-90, mineral-like phase, analcime, Sr-tobermorite, Sr-plagioclase

## Abstract

Co-processing of radioactive effluents with coal fly ash-derived materials is recognized as a resource-saving approach for efficient stabilization/solidification of radioactive components of wastewater. In this context, the paper is focused on the hydrothermal synthesis of Sr^2+^-bearing aluminosilicate/silicate phases as analogs of a mineral-like ^90^Sr waste form using hollow glass-crystalline aluminosilicate microspheres from coal fly ash (cenospheres) as a glassy source of Si and Al (SiO_2_-Al_2_O_3_)_glass_) and Sr(NO_3_)_2_ solutions as ^90^Sr simulant wastewater. The direct conversion of cenosphere glass in the Sr(NO_3_)_2_-NaOH-H_2_O-(SiO_2_-Al_2_O_3_)_glass_ system as well as Sr^2+^ sorption on cenosphere-derived analcime (ANA) in the Sr(NO_3_)_2_-H_2_O-ANA system were studied at 150–200 °C and autogenous pressure. The solid and liquid reaction products were characterized by SEM-EDS, PXRD, AAS and STA. In the Sr(NO_3_)_2_-NaOH-H_2_O-(SiO_2_-Al_2_O_3_)_glass_ system, the hydrothermal processing at 150–200 °C removes 99.99% of the added Sr^2+^ from the solution by forming Sr-tobermorite and Sr-plagioclase phases. In the Sr(NO_3_)_2_-H_2_O-ANA system, Sr^2+^ sorption on analcime results in the formation of solid solutions (Na_1−n_Sr_n/2_)AlSi_2_O_6_·H_2_O of the Na-analcime–Sr-wairakite series. The results can be considered as a basis for the development of environmentally sustainable technology for ^90^Sr removal from wastewater and immobilization in a mineral-like form by co-processing waste from coal-fired and nuclear power plants.

## 1. Introduction

The strategy for sustainable development of power production cannot be realized without solving the environmental problems caused by the formation and accumulation of huge amounts of solid and liquid waste [1]. In particular, coal fly ash (CFA) [2,3,4] and liquid radioactive waste [5] have arisen from the operation of coal-fired and nuclear power plants, respectively. Co-processing of radioactive effluents with CFA-derived materials is recognized as a resource-saving and cost-effective approach for CFA minimization and the efficient removal and stabilization/solidification of radioactive components of wastewater [6,7,8]. In most cases, as-produced multicomponent fly ashes of appropriate bulk chemical composition are used as cementitious materials to produce Portland cement- [9] and geopolymer-based [3,10,11] concrete waste forms or as source materials to synthesize zeolites entrapping radionuclides from wastewater due to their efficient ion exchange properties [12,13,14,15].

The environmentally safe management of high-level radioactive waste of all types is oriented at concentrating and confining the most hazardous radionuclides (^137^Cs, ^90^Sr or long-lived actinides) in chemically, thermally and radiation-resistant solid matrices, such as crystalline and glass-crystalline ceramic materials, in which radioactive elements accommodate in host phases and are similar to naturally occurring minerals in their composition and structure [5,16,17]. This is the most preferred option for hazardous waste disposal compared to other immobilization technologies that use inorganic binders [9,11]. In the context of immobilizing radionuclide ^137^Cs and ^90^Sr, the largest contributors to nuclear waste activity, the mineral-like phases of Cs-feldspathoids (e.g., pollucite) and Sr-feldspars (e.g., plagioclase) types are of a great practical value [17,18]. Following the principles of industrial ecology, the tailoring of cesium or strontium aluminosilicate-based ceramics starting from the CFA-derived precursor of required chemical and mineral-phase composition can be considered a promising method of ^137^Cs and ^90^Sr stabilization and disposal.

Numerous studies have shown that there are valuable microsphere products in multicomponent fly ashes [2,19], which have great potential as precursors of functional materials for various high-tech applications [20,21,22]. Among them, hollow aluminosilicate glass-crystalline microspheres, or cenospheres, are considered to be one of the most important value-added components of sialic CFAs [23,24,25,26,27]. Advanced utilization strategies require significant CFA cenosphere processing to extract the close-cut products with stabilized physical characteristics (size, density, magnetic properties, etc.), composition and globule structure. At present, the fine classification of cenospheres was realized [28,29,30], resulting in cenosphere fractions of the specific chemical, phase and granulometric composition with predictable high-tech properties [31,32,33,34]. 

The availability of close-cut cenosphere products with the (SiO_2_/Al_2_O_3_)_wt._ ratios in the range of 1.2–3.5 was a key factor to use them as precursors for the synthesis of ^137^Cs/^90^Sr mineral-like host compounds (pollucite, Sr-feldspar). To achieve this, different approaches were used, such as (i) impregnation of loose cenospheres or cenosphere-based open-cell porous materials with Cs^+^/Sr^2+^ salts followed by calcination of blends at 800–1100 °C [35,36], (ii) hydrothermal conversion of cenospheres into NaP zeolite followed by Cs^+^/Sr^2+^ sorption and heating of the Cs^+^/Sr^2+^ loaded zeolites up to 1100 °C [37] and (iii) direct synthesis of mineral-like aluminosilicate phases under hydrothermal conditions using cenospheres as a glassy Si and Al source [38,39]. 

The cenosphere-based hydrothermal synthesis seems to be the promising co-processing route to ^137^Cs and ^90^Sr immobilization in mineral-like forms because it can be implemented via the one-step mild thermobaric treatment of the Cs^+^/Sr^2+^-bearing alkaline solution-cenospheres mixture. As reported [38,39], in the Cs^+^-containing system, the pollucite–analcime solid solutions have been easily synthesized in a wide range of cesium concentrations at 150 °C using narrow cenosphere fractions with a high content of a glass phase (≥90 wt.%). Under applied hydrothermal conditions, the efficiency of trapping Cs^+^ from simulant solutions was up to 96%. At the same time, the hydrothermal co-processing of Sr^2+^-bearing solutions and cenospheres has not been studied yet in detail. 

The paper presents the results of studying the hydrothermal conversion of cenospheres of stabilized composition and cenosphere-derived material into the Sr^2+^-bearing mineral-like phases using a strontium stable isotope as an imitator of ^90^Sr. Two experimental approaches were applied, (i) a one-step hydrothermal crystallization of Sr^2+^-bearing phases in the Sr(NO_3_)_2_-NaOH-H_2_O-(SiO_2_-Al_2_O_3_)_glass_ system at temperatures not higher than 200 °C and (ii) a two-step process based on the hydrothermal conversion of cenospheres into analcime followed by the Sr^2+^ sorption on the cenosphere-derived analcime under hydrothermal conditions at elevated temperatures. This methodology was expected to provide both Sr^2+^ removal from the reaction solution and incorporation in crystalline silicate and/or aluminosilicate phases as solid precursors of final ceramic waste forms suitable for ultimate safe disposal.

## 2. Materials and Methods

### 2.1. Chemicals and Materials

Sodium hydroxide (“p.a.”, Vekton, Russia) and strontium nitrate (“puriss.”, LenReactive, Russia) were used for the preparation of non-radioactive Sr^2+^-bearing solutions.

The cenosphere fraction with ~95 wt.% glass phase of (SiO_2/_Al_2_O_3_)_wt._ = 3.1 (marked as (SiO_2_-Al_2_O_3_)_glass_) was the product of the separation of a CFA cenosphere concentrate resulting from the combustion of Kuznetsk coal (Russia) according to the reported procedures [28,29,30]. Chemical and phase compositions (wt.%) of the initial cenosphere fraction were as follows [40]: SiO_2_—67.6, Al_2_O_3_—21.0, Fe_2_O_3_—3.2, CaO + MgO + Na_2_O + K_2_O—7.7; quartz—3.4, mullite—0.8, calcite—0.5, glass phase—95.4; (SiO_2_/Al_2_O_3_)_glass_—3.1. Micrographs of the cenosphere globules are given in Figure 1a.

### 2.2. Hydrothermal Procedures

#### 2.2.1. Hydrothermal Crystallization

Reaction mixtures with a liquid-to-solid (L/S) ratio of about 7/1 (*v*/*v*) were prepared by the addition of cenospheres (~5.0 g) and Sr(NO_3_)_2_ to 1.5 M NaOH resulting in two Sr(NO_3_)_2_-NaOH-H_2_O-(SiO_2_-Al_2_O_3_)_glass_ systems with different molar compositions and Sr/Si ratios (Table 1). The reaction mixture was transferred into a Teflon-lined stainless-steel autoclave (“Beluga”, Premex AG, Switzerland) which was tightly closed without preliminary evacuation. The syntheses were carried out at 150, 180 and 200 °C with autogenous pressure for 24 h under stirring at a rate of 50 rpm. The solid products were separated by filtration, washed with distilled water several times until neutral reaction of a supernatant occurred and isolated by centrifuging the suspension. The sediments were dried at 80 °C in air and the filtrates were analyzed for Sr^2+^ content. The products thus prepared are denoted as xSr-T hereafter, where x is a Sr/Si atomic ratio in the initial mix and T is temperature of treatment.

#### 2.2.2. Hydrothermal Sorption

Analcime was synthesized at 150 °C from the cenosphere fraction according to the reported procedure [38] and used as a sorbent (ANA). Part of the analcime material was hydrothermally treated at 200 °C for 24 h in order to follow possible structure alterations. The typical sorption experiment was as follows. About 1 g ANA was contacted with a 100 mL Sr(NO_3_)_2_ solution of a given Sr^2+^ concentration, such as 500 and 1000 mg/L, at 25, 150 and 200 °C for 24 h in the autoclave under stirring at a rate of 50 rpm. Then the sorbent was separated by filtration, washed with distilled water and dried at 80 °C in air. The products are denoted as xSr/ANA-T, where x is Sr^2+^ concentration in the initial Sr(NO_3_)_2_ solution and T is temperature of treatment (Table 2). Micrographs of the analcime-bearing globules are shown in Figure 1b.

### 2.3. Characterization Techniques

Powder X-ray diffraction (PXRD) data were collected on DRON-3 (IC “Bourevestnik”, St. Petersburg, Russia) and X’Pert PRO (PANalytical, Almelo, Netherlands) diffractometers equipped with a solid-state detector PIXcel using Cu Kα radiation (2θ range 12–120°). The samples were prepared by grinding with octane in an agate mortar and packed into a flat sample holder for the PXRD measurements in Bragg–Brentano geometry. The full-profile crystal structure analysis was performed using the Rietveld method [41] with derivative difference minimization (DDM) [42] refinement. The Na/Sr ratio was estimated from the refined occupancy of respective atomic sites in the analcime lattice taking into account the substitution of two Na^+^ ions by one Sr^2+^. An accurate crystal structure model of analcime obtained in [43] was utilized in the DDM refinement. The crystallographic database of the Joint Committee on Powder Diffraction-International Centre for Diffraction Data (JCPDS-ICDD, now known as the ICDD) JCPDS-ICDD PDF-2 Release 2004 and software PhasanX 2.0 were used to process PXRD patterns.

Morphologies of product particles were studied by scanning electron microscopy (SEM) using TM-1000, TM-3000 and TM-4000 (Hitachi, Tokyo, Japan) instruments. To study the elemental composition of surfaces of product particles, energy-dispersive X-ray spectroscopy (EDS, EDX) examination was performed using TM-3000 and TM-4000 microscopes equipped with the microanalysis system (Bruker, Billerica, MA, USA) including an energy-dispersive X-ray spectrometer with an XFlash 430 H detector and QUANTAX 70 software. 

Simultaneous thermal analysis (STA) was performed on a TG-DSC NETZSCH STA 449C (Selb, Germany) analyzer equipped with an Aeolos QMS 403C mass spectrometer. The measurements were carried out under dynamic 20% O_2_-Ar atmosphere at ambient pressure on heating in the range of 40–600 °C in Pt crucibles with perforated lids (a sample mass 10–12 mg; β = 10°/min). The qualitative composition of the gas phase was determined by on-line QMS in the Multiple Ion Detection mode from the intensity of ions *m*/*z* = 18 (H_2_O), 32 (O_2_) and 44 (CO_2_).

The Sr^2+^ concentration in the post-synthesis solutions was determined by the atomic absorption analysis at an AAS-30 spectrophotometer (Carl Zeiss, Jena, Germany).

The sorbed Sr^2+^ specific quantity in the solid phase (Q_s_, mg/g) was determined as Q_s_ = (C_o_ − C)·*v*/*m*, where C_o_ and C are Sr^2+^ concentrations (mg/L) in the initial liquid phase and post-synthesis liquor, accordingly; V (L) is the volume of solution; and m (g) is mass of the specimen.

The efficacy of Sr^2+^ trapping from the reaction solutions (E, %) was calculated according to the equation E = [(m_o_ − m)/m_o]_ × 100%, where m_o_ and m are the Sr^2+^ quantities (mg) in the initial and post-synthesis solutions, respectively.

## 3. Results and Discussion

### 3.1. Hydrothermal Crystallization

As seen from Table 1, the hydrothermal processing of all the Sr(NO_3_)_2_-NaOH-H_2_O-(SiO_2_-Al_2_O_3_)_glass_ systems provides the near quantitative, no less than 99.99%, Sr^2+^ trapping from reaction solutions into solid products.

Figure 2 shows PXRD patterns of products resulted from crystallization of cenosphere glass at different temperatures in the presence of a varied strontium amount. By the PXRD data, phases of cubic analcime, NaAlSi_2_O_6_·H_2_O (ANA, ICDD 01-070-1575), orthorhombic or triclinic tobermorite, Ca_2.25_Si_3_O_7.5_(OH)_1.5_·H_2_O (ICDD 04-011-0271; 04-014-8455) and feldspar, such as plagioclase, Na_0.499_Ca_0.499_K_0.031_Al_1.488_Si_2.506_O_8_ (WWW-Mincryst, 3714) or Ca_0.65_Na_0.35_Al_1.65_Si_2.35_O_8_ (ICDD 01-083-1369), were identified. The definite identification of tobermorite and plagioclase phases is rather complicated in the polyphase system because of overlapping low-intensive peaks. Cubic analcime was the dominant phase among the crystal phases formed. The temperature rise and the Sr content increase in the reaction mixtures resulted in an intensity increase of the PXRD peaks related to tobermorite and plagioclase (Figure 2a,b). This observation makes it obvious that strontium is involved in the formation of these phases. A slight shift in the position of the PXRD peaks compared to the reference phases not containing strontium also gives reason to propose the formation of (Sr,Ca)-bearing tobermorite and plagioclase with participation of calcium being part of cenosphere glass and added strontium (Figure 3e–h and Figure 4e–h).

Microstructures of the product particles with a visible amount of tobermorite and plagioclase phases synthesized at 150 and 200 °C are presented in Figure 3 and Figure 4 by SEM data. It can be seen that the products consist of devitrified hollow microspheres, and their chips and particles have an irregular shape with inhomogeneous composition and structure. Typically, the majority of microspheres have a multilayered composite shell with compact or loose analcime [44] covering (Figure 3a,b,d and Figure 4a,d) and underlying Sr-bearing matters (Figure 3b–h). Analcime crystals can be found also in the internal void of microsphere globules. The strontium content in analcime is insignificant and amounts to 3.0–5.0 wt.% corresponding to the range of Sr/Si molar ratio of 0.04–0.08. Analcime crystals are supported by porous matter enriched with strontium (Sr/Si = 0.3–0.5) and composed of fine-needle and hair-like formations as well as residues of unconverted glass (Figure 3).

Individual Sr enriched (Sr/Si ~0.7) globules not coated by analcime were also revealed among the product particles (Figure 4b). In addition, the prismatic elongated crystals of 8–10 μm in length being typical for feldspars [45] and containing about 42 wt.% Sr were disseminated in the product matter (Figure 3c and Figure 4d,f,h). There are radial aggregates (Sr/Si = 0.4) with a needle-shaped habit of crystals in the voids of broken globules, which is characteristic of tobermorite (Figure 4c,e) [46]. According to SEM-EDS data, the needle-like tobermorite crystals contain 30–35 wt.% Sr and 3–6 wt.% Ca (Figure 4g) confirming the formation of Ca,Sr-bearing solid solutions (Ca,Sr)_2.25_Si_3_O_7.5_(OH)_1.5_·H_2_O.

Thus, based on the PXRD, SEM-EDS and AAS characterizations of solid products and post-synthetic solutions, it was established that the hydrothermal treatment of the Sr(NO_3_)_2_-NaOH-H_2_O-(SiO_2_-Al_2_O_3_)_glass_ systems at 150–200 °C removes 99.99% of added strontium from the solution by forming Sr enriched mineral-like phases, such as (Sr,Ca)-tobermorite and (Sr,Ca)-plagioclase.

### 3.2. Hydrothermal Sorption

As demonstrated above, the direct hydrothermal co-processing of Sr(NO_3_)_2_ solutions and cenospheres in an alkaline medium resulted in the crystallization of Sr-bearing silicate phases, such as analcime, (Sr,Ca)-tobermorite and (Sr,Ca)-plagioclase, among which the analcime phase incorporates a negligible Sr amount in all range of Sr^2+^ content in the systems. At the same time, pollucite–analcime solid solutions (Na_n_Cs_1−n_)AlSi_2_O_6_·nH_2_O were easily crystallized as the main host phases for cesium in the similar co-processing route [38,39]. Moreover, Komarneni et al. [47] reported about the formation of analcime-based Sr-wairakite, SrAl_2_Si_4_O_12_·2H_2_O, when Sr(OH)_2_ interacts with clinoptilolite and clay under more prolonged thermobaric treatment at 200 °C and 30 MPa. With this connection noted, it was of interest to obtain the Sr^2+^-enriched analcime-based phases under milder conditions (25–200 °C, autogenous pressure). The sorption approach was applied to incorporate Sr^2+^ in the analcime lattice by means of ion exchange at elevated temperatures and pressures with the use of sodium-bearing analcime (Figure 1b) displaying poor Sr^2+^ capacity at 25 °C [48] but capable of trapping Sr^2+^ under hydrothermal conditions at 250–300 °C [49].

Table 2 shows the Sr^2+^ sorption value and efficacy of Sr^2+^ removal upon contacting the analcime-based sorbent with Sr(NO_3_)_2_ solutions at 25, 150 and 200 °C. It is evident that the degree of Sr^2+^ removal increases markedly as temperature rise but does not exceed ~92% for the lower Sr^2+^ concentration (500 mg/L). In turn, the Sr^2+^ sorption capacity is appropriately greater at the higher Sr^2+^ concentration (1000 mg/L) in all the temperature ranges.

The SEM-EDS data for hydrothermally treated analcime in Figure 5 point to the fact that the formation of Sr^2+^ silicate compounds attached to the analcime surface (Figure 5a,b—area b-1,e,f—area f-1) and Sr^2+^ enrichment of the analcime phase area (Figure 5b—area b-2,f—area f-2) take place.

In other words, Sr^2+^ cations are trapped from the solution by two ways, (i) Na^+^ exchange for Sr^2+^ in the analcime structure and (ii) interaction of Sr^2+^ cations with silicate species occurring in the porous structure of the analcime-bearing material as a result of its preliminary synthesis from aluminosilicate glass with (SiO_2_/Al_2_O_3_)_glass_ > (SiO_2_/Al_2_O_3_)_ANA_ (Figure 1c–f) [38].

As revealed, the Sr^2+^ content in the analcime phase of the sorbent material increases regularly as the temperature rises up to 150 °C and 200 °C for the same initial Sr^2+^ concentration in the liquid phase. The EDX spectra for local parts of the analcime layer after contacting with strontium nitrate solutions at 25 °C (500Sr/ANA-25) and 200 °C (1000Sr/ANA-200) are presented in Figure 5d,h, respectively. Compositions of Na,Sr-ANA phases calculated by SEM-EDS data satisfy the conode NaAlSi_2_O_6_–Sr_0.5_AlSi_2_O_6_ in a ternary diagram Na-Sr-Al (Figure 6) that corresponds to the formation of solid solutions (Na_1−n_Sr_n/2_)AlSi_2_O_6_·H_2_O of the Na-analcime–Sr-wairakite series. Based on the data obtained, one can assume that, because of sorption runs, the solid solutions (Na_1−n_Sr_n/2_)AlSi_2_O_6_·H_2_O being part of the Na-analcime–Sr-wairakite series are formed.

The PXRD analysis with DDM refinement was used to monitor the variation of the analcime lattice parameter, which is sensitive to the substitution of Na^+^ ions by Sr^2+^ in the lattice, and to estimate the composition of Na,Sr-ANA phases. Observed and calculated PXRD patterns after the DDM refinement for samples ANA and 1000Sr/ANA-200 are presented, as an example, in Figure 7.

Table 2 summarizes the composition and cubic (*Ia*-3*d*) lattice parameters for initial ANA phases and Na,Sr-ANA phases obtained from both the ICDD database and full-profile structure analysis. As seen from Table 2, the Sr^2+^ sorption at 25 °C does not cause statistically significant changes in values of the lattice parameters compared to the untreated analcime. Due to these facts, the assumption was made that the Sr^2+^ sorption on analcime at 25 °C was negligibly low, and detection of strontium in the analcime phase by SEM-EDS was likely to be caused by the Sr^2+^ sorption on alternative sorption sites of the inhomogeneous analcime-bearing material, for example, residues of unreacted glass supporting the analcime crystals (Figure 1b,d) [38]. As seen in Figure 5, a significant part of sorbed Sr^2+^ can be localized in the Sr-enriched compounds providing the observed degree of Sr^2+^ removal from the liquid phase at 25 °C (Table 2). The marked changes of the lattice parameters are observed only for analcime phases exposed to the hydrothermal treatment in the presence of Sr^2+^ at 150 °C and 200 °C. The full-profile crystal structure refinement for the Na,Sr-analcime phases resulted from Sr^2+^ sorption at elevated temperatures allowed the composition of solid solutions (Na_1−n_Sr_n/2_)AlSi_2_O_6_·H_2_O formed at these temperatures (Table 2). As shown in Figure 8, the Sr^2+^ content in the Na,Sr-ANA solid solutions calculated from the PXRD data (Q_xrd_) correlates with the sorbed Sr^2+^ quantity (Q_s_) in the specimen determined by the AAS data of filtrate solutions. The linear dependence does not go through a zero point at the *x*-axis and intercepts the segment at ~16 mg/g. That is in agreement with the existence of at least two forms of the Sr^2+^ binding upon contacting analcime with Sr(NO_3_)_2_ solutions, and the formation of the Sr compound is responsible for the shift of the linear correlation.

Additional indication of formation of solid solutions (Na_1−n_Sr_n/2_)AlSi_2_O_6_·H_2_O is the result of simultaneous thermal analysis. It is known that when Na^+^ cations are substituted for Sr^2+^, the Sr^2+^ ions are placed in the same position as the analcime lattice [44]. In this case, the water content in its structure would not be changed, but the binding energy of water molecules with the aluminosilicate framework can lessen due to the decrease of the number of two-charged cations. To estimate the water state in the analcime structure, thermogravimetric analysis (TG/DTG) is sufficiently informative.

The heating of all samples in the range of 40–600 °C was accompanied by continuous mass loss up to temperatures of about 450 °C. By the MS analysis of off-gases, this loss is caused by the water evolving which occurred as two slightly resolved steps at 40–150 °C and 150–450 °C. The analcime samples, before and after hydrothermal treatment in the presence of strontium salt, showed similar behavior in heating with the water content being approximately equal (Table 2). At the same time, the increase in Sr content in analcime resulted in narrowing peaks and shifting the peak maxima to low temperatures. This observation points to a reduction of the binding energy of water molecules and a more uniform distribution by dehydration enthalpy. The effect becomes the most apparent for samples after sorption at 200 °C, which are characterized by the highest degree of substitution of Na^+^ for Sr^2+^ cations (Table 2).

Thus, as a result of the implementation of the sorption approach, including the Sr^2+^ sorption on analcime under hydrothermal conditions at 150–200 °C, the formation of solid solutions (Na_1−n_Sr_n/2_)AlSi_2_O_6_·H_2_O of the Na-analcime–Sr-wairakite series was demonstrated. The elevated temperatures favor the Sr^2+^ incorporation in the analcime-based phases with the simultaneous increase of the Sr^2+^ removal efficiency from Sr^2+^-containing solutions.

## 4. Conclusions

The hydrothermal treatment of strontium-bearing solutions in the presence of coal fly ash cenospheres and cenosphere-derived zeolite material was demonstrated as a model for sustainable co-processing of wastes generated by coal-fired and nuclear power plants in order to remove the radioactive contaminant from wastewater and fabricate a mineral-like waste form for its final disposal. Two experimental approaches were implemented, (i) the direct hydrothermal synthesis of Sr-bearing mineral-like phases (Sr-tobermorite, Sr-plagioclase) in an alkaline medium using cenospheres as a glassy source of Si and Al at temperatures not higher than 200 °C and (ii) the Sr^2+^ sorption on cenosphere-derived analcime under the same hydrothermal conditions resulting in solid solutions (Na_1−n_Sr_n/2_)AlSi_2_O_6_·H_2_O of the Na-analcime–Sr-wairakite series. Among these two methods, hydrothermal synthesis seems to be more efficient because it has the highest degree (>99.99%) of strontium removal from solutions. The fabrication of Sr-bearing matrix materials is carried out in a hermetically sealed autoclave and characterized by both low temperature (no more than 200 °C) and low emission.

The results were obtained jointly with the ability of cenospheres to bind cesium (I) in the structure of pollucite–analcime solid solutions even at 150 °C which can be considered as original data for the development of resource-saving and energy-efficient hydrothermal technology for ^137^Cs and ^90^Sr removal from wastewater and immobilization in a mineral-like form. A wide variety of ^137^Cs and ^90^Sr-bearing aqueous radioactive wastes can be the object of its application including high sodium acid and alkaline radioactive wastes resulting from spent nuclear fuel processing and fast neutron reactors decommissioning, accordingly.

Another potential of cenospheres and cenosphere-based analcime in relation to the problem of radioactive waste disposal can be connected with the possibility to use the materials in underground nuclear waste repositories as a sorptive barrier for migrating radionuclides. Elevated temperatures, up to 250–300 °C, in a near surrounding field create suitable conditions for the hydrothermal processing of soluble ^137^Cs and/or ^90^Sr leached from the nuclear waste form with underground water.

## Figures and Tables

**Figure 1 materials-14-05586-f001:**
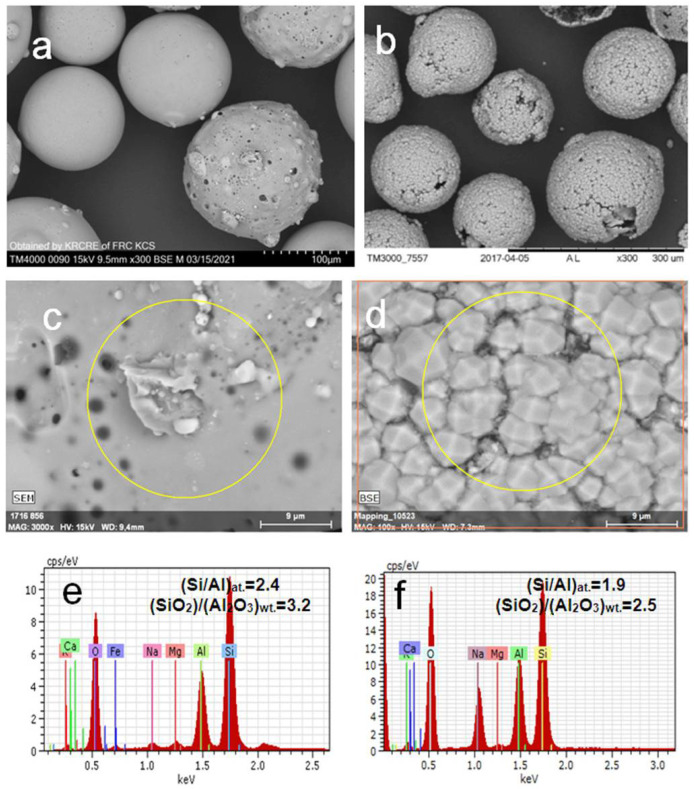
SEM images of (**a**) initial cenosphere material and (**b**) analcime-bearing globules; BSE micrographs and associated EDX spectra for local parts of (**c**,**e**) cenosphere surface and (**d**,**f**) analcime layer.

**Figure 2 materials-14-05586-f002:**
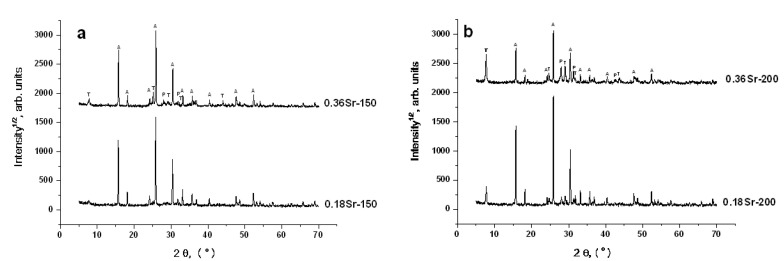
PXRD patterns of products resulting from hydrothermal crystallization in the Sr(NO_3_)_2_-NaOH-H_2_O-(SiO_2_-Al_2_O_3_)_glass_ systems at (**a**) 150 °C and (**b**) 200 °C: A—analcime; T—tobermorite; P—plagioclase.

**Figure 3 materials-14-05586-f003:**
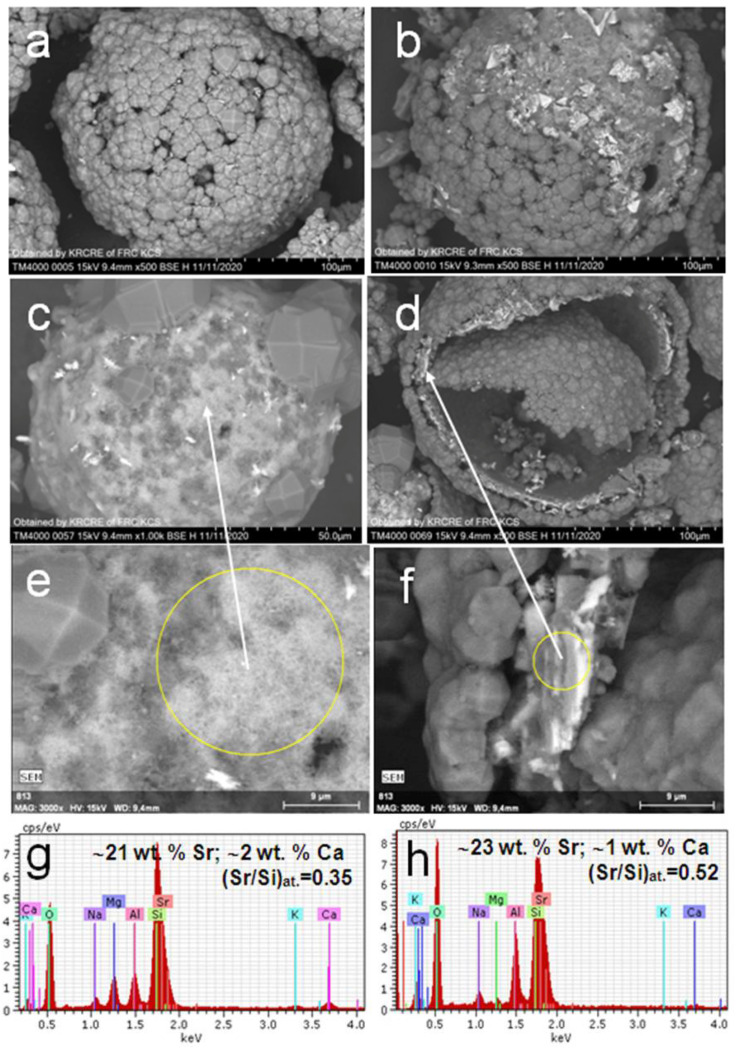
SEM images of product particles resulting from (**a**–**d**) hydrothermal crystallization in the Sr(NO_3_)_2_-NaOH-H_2_O-(SiO_2_-Al_2_O_3_)_glass_ systems at 150 °C; (**e**–**h**) BSE micrographs and associated EDX spectra for local parts of the particles.

**Figure 4 materials-14-05586-f004:**
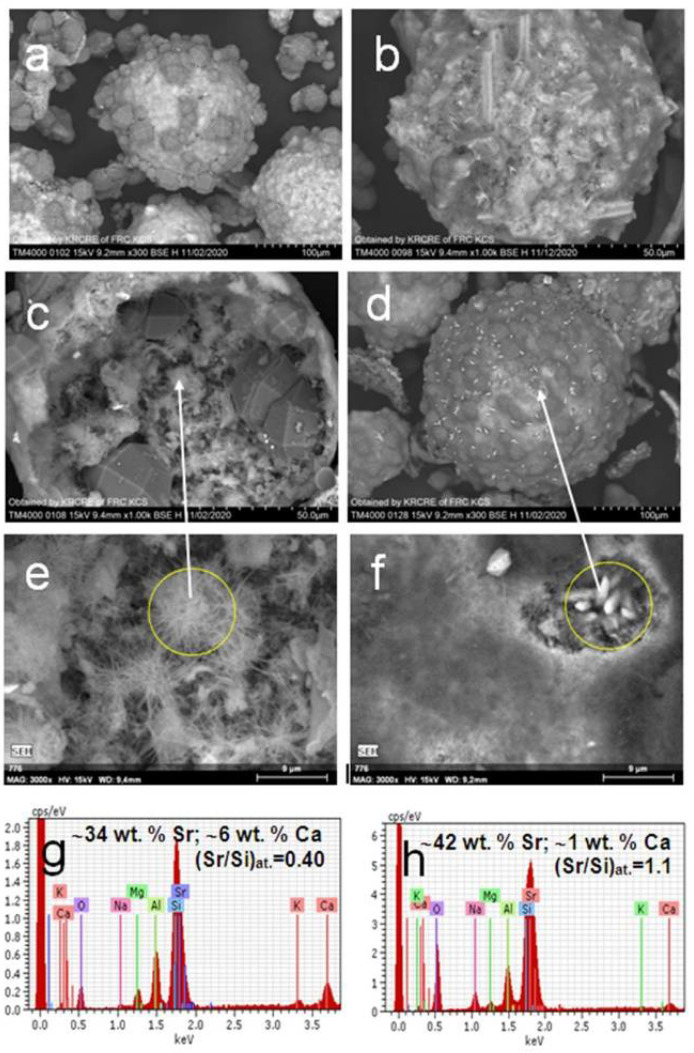
SEM images of (**a–d**) product particles resulting from hydrothermal crystallization in the Sr(NO_3_)_2_-NaOH-H_2_O-(SiO_2_-Al_2_O_3_)_glass_ systems at 200 °C; (**e**,**f**) BSE micrographs and (**g**,**h**) associated EDX spectra for local parts of the particles.

**Figure 5 materials-14-05586-f005:**
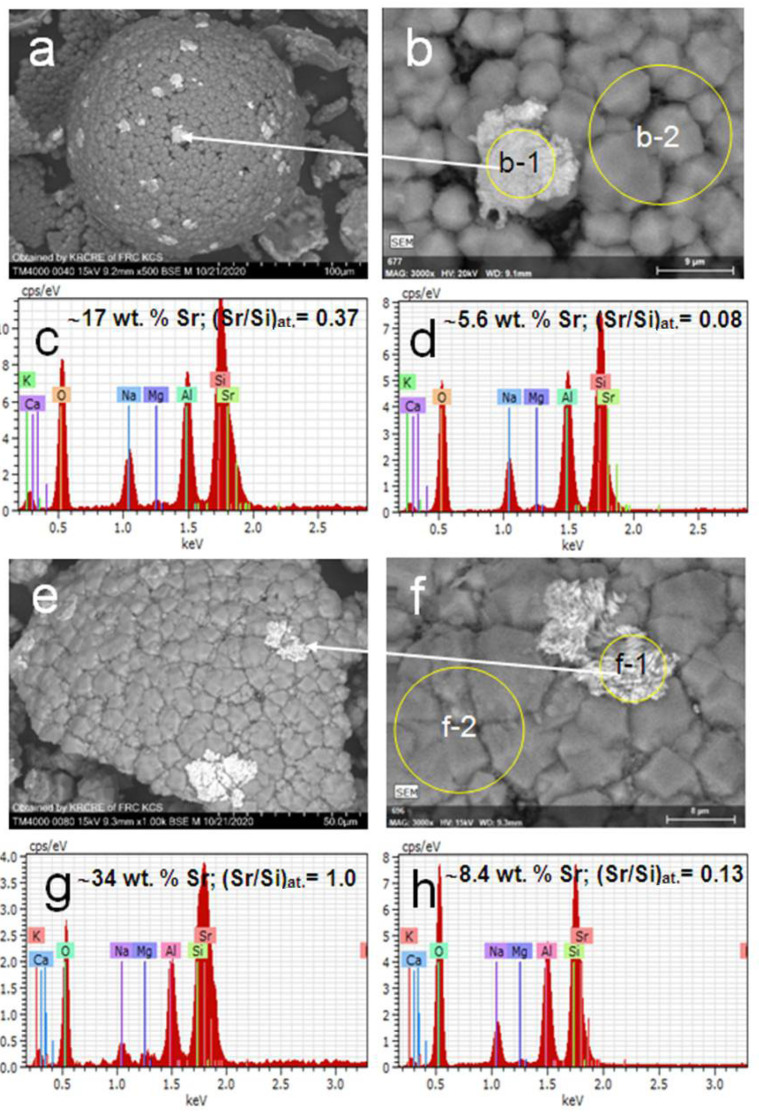
SEM images of product particles resulted from hydrothermal treatment of the Sr(NO_3_)_2_-H_2_O-ANA systems at (**a**) 25 °C and (**e**) 200 °C; (**b**,**f**) BSE micrographs and associated EDX spectra for local parts of the particles: (**c**,**d**)—for points b-1 and b-2, accordingly; (**g**,**h**)—for points f-1 and f-2, accordingly.

**Figure 6 materials-14-05586-f006:**
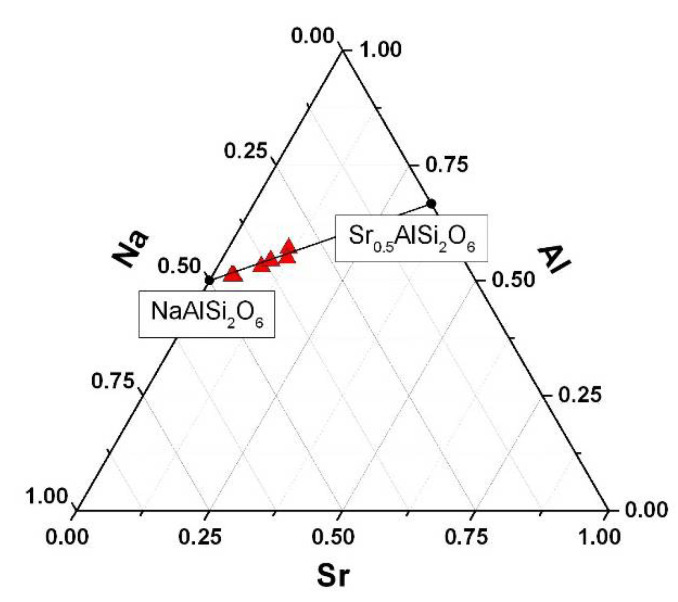
Composition of solid solutions (Na_1−n_Sr_n/2_)AlSi_2_O_6_·H_2_O in Na-Sr-Al coordinates (at.%) by SEM-EDS data.

**Figure 7 materials-14-05586-f007:**
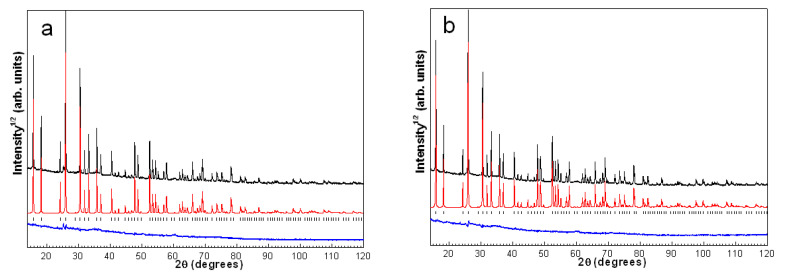
Observed (top, black), calculated (mid, red) and difference (bottom, blue) PXRD profiles after DDM crystal structure refinement for samples (**a**) ANA and (**b**) 1000Sr/ANA-200.

**Figure 8 materials-14-05586-f008:**
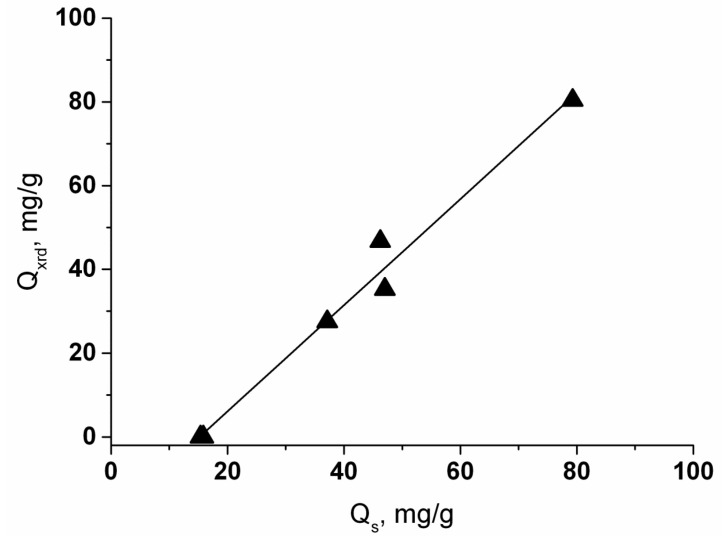
The dependence of the Sr^2+^ content in analcime by XRD data (Q_xrd_) on the sorbed Sr^2+^ quantity (Q_s_) in the specimen by the AAS data of filtrate solutions.

**Table 1 materials-14-05586-t001:** Molar composition of reaction mixtures and efficacy of the Sr^2+^ trapping from reaction solutions of Sr(NO_3_)_2_-NaOH-H_2_O-(SiO_2_-Al_2_O_3_)_glass_ systems.

System	Sample	T, °C	E, %
5.5SiO_2_/1.0Al_2_O_3_/1.0SrO/3.65Na_2_O/270H_2_O (Sr/Si)_at._ = 0.18	0.18Sr-150	150	99.99
	0.18Sr-180	180	99.99
	0.18Sr-200	200	99.99
5.5SiO_2_/1.0Al_2_O_3_/2.0SrO/3.65Na_2_O/270H_2_O (Sr/Si)_at._ = 0.36	0.36Sr-150	150	>99.99
	0.36Sr-180	180	>99.99
	0.36Sr-200	200	99.99

**Table 2 materials-14-05586-t002:** Composition, lattice parameters for Na,Sr-ANA phases, Sr^2+^ sorption and efficacy of Sr^2+^ extraction from solutions, TG data for the product.

Sample	Composition	Lattice Parameter, Å	Sr^2+^ Sorption, mg/g	E, %	Δm ^3^, % (40–150 °C)	T_m_, ^4^ °C
ANA	NaAlSi_2_O_6_·H_2_O ^1^	13.7332 (1)	-	-	7.77 (0.65)	302
1000Sr/ANA-25	NaAlSi_2_O_6_·H_2_O ^1^	13.7337 (8)	15.9	22.7	7.89 (0.84)	299
500Sr/ANA-25	NaAlSi_2_O_6_·H_2_O ^1^	13.7339 (8)	15.4	31.7	7.80 (0.79)	300
1000Sr/ANA-150	Na_0.82_Sr_0.09_AlSi_2_O_6_·H_2_O ^2^	13.7288 (6)	47.0	47.7	7.81 (0.68)	295
500Sr/ANA-150	Na_0.86_Sr_0.07_AlSi_2_O_6_·H_2_O ^2^	13.7298 (5)	37.1	72.7	7.74 (0.67)	303
1000Sr/ANA-200	Na_0.58_Sr_0.21_AlSi_2_O_6_·H_2_O ^2^	13.7168 (9)	79.3	80.0	7.77 (0.55)	291
500Sr/ANA-200	Na_0.76_Sr_0.12_AlSi_2_O_6_·H_2_O ^2^	13.7241 (8)	46.2	91.7	7.78 (0.54)	305

^1^ ICDD 01-070-1575. ^2^ By data of the full-profile crystal structure refinement. ^3^ Mass loss under heating at 10°/min in the range of 150–600 °C. ^4^ Temperature of mass loss at a maximal rate (according to the DTG curve).

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
