# Peer review of "Hydrothermal Co-Processing of Coal Fly Ash Cenospheres and Soluble Sr(II) as Environmentally Sustainable Approach to Sr-90 Immobilization in a Mineral-like Form"

_materials, 2021, doi:10.3390/ma14195586_

Round 1

Reviewer 1 Report

I recommend the publication of this work. However, some issues should be addressed before publication:

  1. The presentation of some research results should be improved. It is mainly about tables 1 and 2. In my opinion, they should not contain the results, since they can be found in chapter 2: Materials and Methods. The reference to the content of these tables is on pages 9 and 11 etc., which is not conducive to the clarity and consistency of the text. The test results (i.e. E,% etc.) should be presented in the next chapter: Results and discussion.
  2. Explanations of symbols and abbreviations should appear as soon as they appear in the text (e.g. ICDD etc).
  3. The statements in the chapter Conclusions are too general and should be extended.
  4. The authors have conducted and described a lot of research, but in the article it does not translate sufficiently to the practical conclusions resulting from the achieved results.
  5. English should be improved (especially grammar).

Reviewer 2 Report

The paper investigates the hydrothermal conversion of hollow aluminosilicate glass-crystalline microspheres termed cenospheres of stabilized composition and cenosphere-derived material into the Sr-bearing mineral-like phases to produce of final ceramic wasteforms suitable for ultimate safe disposal.

The paper is of both scientific and practical interest, it is concisely written containing all the details needed to understand the results obtained. It provides sound conclusions and can be published practically as is.

The only comments to authors are on lines 16-17 of Abstract, where they might explain the term “cenosphere” because it is not widely used and might be unknown to some readers. Also, here the word “imitators” could be better replaced by “analogues”.
